# Dabigatran in Cerebral Sinus Vein Thrombosis and Thrombophilia

**DOI:** 10.3390/life12070970

**Published:** 2022-06-28

**Authors:** Lukas Kellermair, Matthias W. G. Zeller, Caterina Kulyk, Josef Tomasits, Tim J. von Oertzen, Milan R. Vosko

**Affiliations:** 1Department of Neurology 2, Kepler University Hospital, Med Campus III, 4020 Linz, Austria; lukas.kellermair@kepleruniklinikum.at (L.K.); matthias.zeller@kepleruniklinikum.at (M.W.G.Z.); caterina.kulyk@kepleruniklinikum.at (C.K.); tim.vonoertzen@kepleruniklinikum.at (T.J.v.O.); 2Medical Faculty, Johannes Kepler University, 4040 Linz, Austria; josef.tomasits@kepleruniklinikum.at; 3Department of Laboratory Medicine, Kepler University Hospital, Med Campus III, 4020 Linz, Austria

**Keywords:** cerebral sinus vein thrombosis, dabigatran peak concentration, factor II mutation, factor V Leiden, thrombus dissolution

## Abstract

Background and Purpose: Thrombophilic gene alterations are a major risk factor for cerebral sinus vein thrombosis (CSVT). Up to 30% of all patients with cerebral sinus vein thrombosis (CSVT) are found to have thrombophilic defects such as prothrombin mutation (PTM) or factor V Leiden (FVL). Their repercussions on the plasma levels of dabigatran etexilate are unclear. In this prospective case–control study, we aimed to investigate whether thrombophilia in CSVT has an influence on dabigatran peak-plasma levels. Methods: We monitored 10 patients over 12 months with acute CSVT, genetic thrombophilia with off-label use of dabigatran etexilate 150 mg twice a day and measured dabigatran peak-plasma levels and radiological outcome. We also monitored patients without genetic thrombophilia with dabigatran etexilate 150 mg twice a day and compared the efficiency and dabigatran peak-plasma levels. Results: Patients with homozygote PTM had significantly lower dabigatran peak concentration compared to patients with FVL or the control group (23 ± 4.2 vs. 152.3 ± 27.5 and 159.6 ± 63.08; *p*-value ≤ 0.05) There was no significant difference in dabigatran etexilate plasma levels between the heterozygote PTM group compared to patients with FVL or the control group (*p* = 0.29). There was no correlation between dabigatran peak concentration and delayed thrombus dissolution. Conclusions: Dabigatran peak concentration was stable in patients with heterozygote FVL and heterozygote PTM, but not in homozygote PTM, compared to controls. Genetic screening for thrombophilia in patients after CSVT may be useful to make patient tailored therapeutic decisions regarding oral anticoagulation and may decrease thrombotic events.

## 1. Introduction

Thrombophilia is a casual finding in patients with cerebral sinus venous thrombosis (CSVT) and includes deficiencies in factor II, antithrombin, protein C, protein S or factor V activity. Up to 30% of patients with CSVT are found to suffer from some form of genetic thrombophilia such as prothrombin-mutation (PTM) or Factor V Leiden (FVL) [1,2]. Notably, prothrombin (factor II) mutations are more often found in CSVT than in deep vein thrombosis (DVT) [2], while FVL mutations (or APC-resistance) are more common in DVT compared to CSVT. Overall FVL mutations are found in 25% of patients with a first event of venous thromboembolism (VTE) of any type [2,3].

Heterozygote FVL mutations are mainly found in the white population in Europe and the US with a prevalence of 10–15%. However, the number of homozygote mutation carriers is much lower, with a frequency of with approximately 1 in 5000 of all FVL mutation carriers [3]. The risk of VTE increases five-fold in heterozygotes, and 50-fold in homozygotes [4,5]. In comparison, the prevalence of PTM is much lower and varies from 0.7 to 4% in Europe [6]. Carriers of this mutation present with increased prothrombin levels and have a 2- to 3-fold increased risk of both venous and arterial thrombosis. Like in FVL, homozygosity is rare; however, the risk for recurrent VTE is 30-times higher [4,6,7].

Most patients with CSVT present with multiple known risk factors for VTE such as pregnancy, postpartum condition, malignancy, hormonal contraceptive or hormonal replacement therapy. However, the role of thrombophilic disorders in recurrent VTE is still unknown and genetic testing for thrombophilic disorders is not recommended in recent guidelines for the management of CSVT [8,9].

The risk for recurrent VTE after CSVT has been reported to be between 2 and 4.1 per 100 persons per year [10,11]. Anticoagulation with vitamin K antagonists was the only established treatment until recently when dabigatran etexilate was shown to be similarly safe and effective for preventing recurrent VTEs in patients with CSVT [12]. Dabigatran etexilate is a direct thrombin antagonist and is a proven treatment for preventing strokes in patients with atrial fibrillation [13]. Since its introduction, the use of dabigatran has been expanded to include the treatment and prevention of VTE [14].

Direct oral anticoagulants (DOACs), including the factor II inhibitor dabigatran and factor Xa inhibitors rivaroxaban, apixaban and edoxaban, have many advantages over warfarin in the treatment of VTE; however, data regarding efficiency and safety in thrombophilia are sparse. To our knowledge, data are only available from a handful of studies, including the TRAPS study in 2018. The study reported a higher incidence of stroke and myocardial infarcts in rivaroxaban- compared to warfarin-treated patients with antiphospholipid syndrome. Another double blind randomized trial for dabigatran in general thrombophilia [15] and some case series have reported conflicting results on the efficacy of DOACs in patients with inherited thrombophilia [16,17]. We did not find any longitudinal data of patients with FVL or PTM in CSVT. Hence, there is still a lot of uncertainty regarding the treatment of patients with thrombophilic disorders with direct anticoagulants [18,19].

The aim of our study was to rate the recurrence of VTE events and dabigatran plasma levels in patients with CSVT and FVL and/or PTM treated with dabigatran.

## 2. Methods

### 2.1. Study Procedures

The study was approved by the local ethical review board (Ethik-Kommission Land Oberösterreich; EK-58-17). Patients were subsequently enrolled in the neurological outpatient department of the Med Campus III, Kepler University Hospital Linz. Patients with CSVT were diagnosed using cranial MRI with venous angiography on 1.5 Tesla MRI Scanners. Acute CSVT was defined as newly acquired neurological symptoms up to 10 days before diagnosis. All consecutive patients with acute CSVT were screened for genetic thrombophilia (FVL, PTM, Antithrombin Deficiency, Protein C and S Deficiency) before treatment. The detection of FVL and PTM was based on molecular tests (polymerase chain reaction-PCR and hybridization).

Inclusion criteria were patients with acute CSVT with or without genetic thrombophilia who refused to take standard therapy (vitamin K antagonists) or the treatment with vitamin K antagonists was not feasible (significantly fluctuating or repeated subtherapeutic international normalized ratios (INR)). These patients received off-label treatment with a standard dose of 150 mg dabigatran etexilate twice a day after additional extensive patient information. Patients with genetic thrombophilia (factor V Leiden mutation or prothrombin mutation) and treatment with 150 mg dabigatran etexilate twice a day were included in the cohort (thrombophilia group). Patients without genetic thrombophilia were included in the control group. The control group consisted of patients who received 150 mg dabigatran etexilate twice a day after ischemic stroke due to atrial fibrillation without genetic thrombophilia and three patients with CSVT without genetic thrombophilia. All patients had similar BMI, age and albumin concentration in the plasma.

Before starting the treatment, we controlled concomitant medications for common pharmalogical interactions with dabigatran metabolism according to the European Heart Rhythm Association (EHRA) guidelines (e.g., phenytoin, rifampicin) [20]. Patients with medications with common pharmalogical interactions with dabigatran metabolism were excluded from the study.

In all patients, additive risk factors for VT (previous DVT, immobilization, surgery, malignancy, pregnancy, hormone therapy, nicotine, body mass index [kg/m^2^], recent long-distance travel, age (years), gender) were evaluated. 

Written informed consent was obtained from all participants. Tests and results were performed according to the principles of the Declaration of Helsinki.

### 2.2. Blood Samples

All patients in both cohorts received routine blood laboratory including Creatinine Clearance, liver function parameters and coagulation parameters such as activated partial thromboplastin time (aPTT) and thrombin time (TT). 

Dabigatran peak plasma concentration were measured at baseline, 3 and 24 months. Baseline meant after 48 h of receiving dabigatran etexilate. Blood samples were taken 2 to 4 h after drug intake, respectively. Dabigatran plasma concentration was measured using the INNOVANCE direct thrombin inhibitor assay (DTI) from Siemens as previously described and according to manufacturer’s instructions [21,22,23]. An insufficient dabigatran peak concentration was defined as <50 ng/mL.

### 2.3. Treatment

Patients with CSVT were admitted to either the hyperacute stroke unit or neurological intensive care unit. After initial treatment with either aPTT-guided unfractionated heparin (UFH) or weight-adjusted low molecular weight heparin (LMWH) twice daily, anticoagulant therapy was continued with off-label dabigatran etexilate 150 mg twice a day, regardless of the presence of intracranial hemorrhage (ICH). In two patients with homozygote PTM, therapy with dabigatran was stopped due to low peak plasmatic levels of dabigatran. Additionally, one patient with homozygote PTM also suffered from a recurrent thrombotic event. Therapy was switched to vitamin K antagonist. In both patients, we performed a diagnostic workup for low plasma levels including gastroscopy (bowel disease) and verified medication interactions. In patients with seizures with or without parenchymal lesions, anti-seizure medication was initiated [24,25].

Delayed thrombus dissolution was defined as residual thrombus signs in the MRI after 3 months of anticoagulation. Chronic CSVT was defined as a sign of residual thrombus after 12 months of anticoagulation.

### 2.4. Statistics

All statistical tests were performed using SPSS version 18.0.2 and Prism Graph-Pad version 9.01. 

Assumptions of normal distribution for continuous variables were tested with the Kolmogorov–Smirnov test with Lilliefors correction. Normally distributed continuous variables were compared using the Student’s t- or the Welch’s *t*-test in case of variance heterogeneity (verification with Levene’s test). The exact Mann–Whitney-U test was applied in case of non-normally distributed continuous variables or ordinal variables. 

The correlation between metric variables were calculated by the Bravais-Pearson Correlation Coefficient. Sample size and power calculation was based on preliminary data. A *p*-value < 0.05 was regarded as statistically significant (Power 90%, two-sided Type-I-error 5%). Correction of multiple testing was performed using Dunn’s correction.

## 3. Results

### 3.1. Baseline Characteristics of Study Patients

We included 24 patients with acute CSVT. Out of 24 patients with CSVT, 41% (10) had genetic thrombophilia; 4 patients with PTM, (two homozygote and two heterozygote) and 6 patients with heterozygote FVL. Only 1 patient had homozygote PTM and heterozygote FVL. Of these, 5 patients had more than one thrombophilic mutation (1 patient: homozygote FII and homozygote PAI-1 4G/5G polymorphism; 1 patient: homozygote FII mutation, homozygote PAI-1 4G/5G polymorphism and heterozygote FV mutation (the patient was included in the homozygote PTM-group); 3 patients: heterozygote FV mutation and heterozygote PAI-1 4G/5G polymorphism). (Table 1 and Table 2)

The control group consisted of patients with a clear indication for oral anticoagulation (3 patients with CSVT without genetic thrombophilia and 24 patients with atrial fibrillation). 

All patients reported regular intake of 150 mg dabigatran etexilate twice a day. The mean age was 48.50 years ± 14.9 in the genetic thrombophilia group and 60.36 ± 21.02 years in the control group (*p*-value 0.1).

The clinical presentation of CSVT in our study were: isolated headaches (62.50%), visual impairments (20.83%) and seizure/focal neurological symptoms (16.67%). The localizations of the thrombus were as follows: sinus transversus (71%), sinus sigmoideus (58%), sinus sagittalis superior (37.5%) and vena jugularis (37.5%). In total, 60% of patients had thrombus in more than one vessel.

### 3.2. Dabigatran Peak Concentration and Coagulation Parameters

Out of the 10 patients with CSVT and genetic thrombophilia, 2 had multiple insufficient dabigatran peak concentrations (26 and 20 ng/mL). Peak plasma concentration was measured in two independent blood samples over 4 weeks to exclude measurement errors. Additionally, 1 out of 2 had early recurrent CSVT. Both patients had a homozygote PTM and homozygote PAI 4G/5G polymorphism. Both were switched to a different anticoagulant (vitamin K antagonists) within 30 days after initial CSVT. All other patients showed sufficient dabigatran peak concentrations (>50 ng/mL) with no significant difference between patients with heterozygote PTM or heterozygote FVL and patients without genetic thrombophilia (Figure 1). None of the other patients had recurrence of CVST. There were no significant differences between dabigatran peak concentration between baseline (BL), and Follow-Up 1 (FU 1) and Follow-Up 2 (FU 2) (Figure 2).

Patients with homozygous PTM had significantly lower dabigatran peak concentrations compared to patients with heterozygous FVL or the control group (median 23; range: 6 vs. 155.5; 70.0 and 137.0; 186.0. adjusted *p*-value for both ≤ 0.05). There was no significant difference in dabigatran etexilate plasma levels between heterozygous PTM and patients with FVL or the control group (median 118; range: 34 vs. 155.5; 70.0 and 137.0; 186.0. adjusted *p*-value 0.29).

APTT were not significantly different in all groups. Under treatment with dabigatran etexilate, the thrombin-time was significantly lower in the two patients with homozygous PTM (median: 57.75; range 20.5) compared to the control group (median: 153.0; range: 49.0; adjusted *p*-value 0.004) and patients with FVL (median 167.5; range: 65.0; adjusted *p*-value 0.001). Comparing homozygous PTM and heterozygous PTM showed no significant differences (median 23; range: 6 vs. median: 118 range: 34; adjusted *p*-value 0.33). The dabigatran peak concentration correlated significantly with the thrombin time (*p* 0.002).

### 3.3. Outcome

Delayed thrombus dissolution (>3 months) was observed in 5 patients (1 with homozygote PTM; 4 with heterozygote FVL. Only 1 patient (with homozygous PTM) suffered from recurrent cerebral vein thrombosis while receiving dabigatran. All other patients with genetic thrombophilia did not have radiological signs of recurrent thrombosis up to 1 year after receiving off-label use of dabigatran etexilate 150 mg twice a day. No patients suffered from chronic CSVT (12 months after onset)

There was no correlation between dabigatran peak concentration and delayed thrombus dissolution.

## 4. Discussion

To our knowledge, this is the first study to investigate dabigatran peak concentration in CSVT in patients with genetic thrombophilia. The efficacy and safety of dabigatran etexilate compared to dose adjusted warfarin has been shown recently by Ferro et al. in CSVT [26,27]. These findings were confirmed in systemic reviews also in other direct oral anticoagulants (DOCS) [28,29]. Therefore, many clinicians view DOACS and especially dabigatran etexilate as a viable alternative to vitamin K antagonists, in particular in young and active patients with CSVT. However, up to 30% of patients with CSVT presented with concomitant heritable thrombophilia such as PTM and FVL. Although our knowledge on DOACS in genetic thrombophilia has improved, the efficacy of dabigatran in thrombophilia is still under discussion.

Up until now, there have been no specific recommendations on the treatment of venous thromboembolism in patients with genetic thrombophilia and few studies have investigated the role of DOACs in genetic thrombophilia. Conflicting results in those studies may result in increasing uncertainty in clinicians when treating patients with such conditions [15,16,18].

The latest prospective cohort study showed a similar efficacy in treating VTE with DOACs or vitamin K antagonists in patients with genetic thrombophilia [17]. However, only 16 patients received dabigatran and data were pooled with other DOACs. Furthermore, only 1 patient in this study was diagnosed with a homozygote factor II mutation and the type of DOAC in this patient is not reported. The RE-COVER/RE-COVER II and RE-MEDY studies did not report any issues regarding dabigatran etexilate treatment in the presence of thrombophilia; however, the authors did not distinguish between heterozygote and homozygote variants of factor V Leiden and factor II mutation [15].

A systematic review by Khider et al. in 2022 showed encouraging results in the efficacy and safety of DOACS in genetic thrombophilia. However, again, only a few cases of patients with homozygote factor II mutation were reported and in no patients were dabigatran peak plasma concentrations tested [30].

Similar results have been published in a study investigating DOAC treatment in antiphospholipid syndrome (APS) [31]; however, conversely, in 2019, a systematic review reported a recurrent increased thrombosis rate during DOAC treatment compared to warfarin in APS [32], which led to a “Dear Doctor Letter” and the stop of usage of DOACs in APS.

Interestingly, our study on patients with genetic thrombophilia showed similar dabigatran etexilate concentrations compared to our control group, except for patients with homozygous PTM. The dabigatran peak concentration and thrombin time (TT) were significantly lower in these patients compared to heterozygous FVL and patients without genetic thrombophilia. Although TT is not a good marker to quantify dabigatran peak concentrations, low TT suggests a lack of dabigatran induced anticoagulation in these patients [33]. TT is usually unmeasurable by automated coagulation analyzers, even at low dabigatran concentrations [34].

To exclude other reasons for the observed lack of dabigatran action, we performed gastroscopy to exclude gastritis and checked for potential pharmacological interactions.

To our knowledge, it is unclear why PTM leads to insufficient dabigatran peak concentrations. This genetic mutation results in inherently increased plasma prothrombin levels and resulting increased plasma thrombin levels. Since the action of dabigatran is a result of its binding to prothrombin and thrombin, elevated levels of those proteins might significantly exceed the number of dabigatran molecules available under the usual dabigatran dosage scheme. Interestingly, a study reported, that prothrombin complex concentrates may be able to reverse dabigatran-induced anticoagulation in a dose-dependent manner due to the elevation of thrombin plasma levels to surpass that of dabigatran etexilate [35,36]. Hence, higher dabigatran dosages may be necessary in homozygous PTM patients to counteract the elevated number of binding partners and ensure adequate anticoagulation.

Our study has limitations. First and foremost, our sample size is very small. In our consecutive cohort, 2 patients with homozygous PTM, 2 patients with heterozygote PTM and 6 patients with FVL were studied. Although heterozygous PTM did not have significantly lower plasma levels of dabigatran etexilate or thrombin time compared to our control group, the results were also not significant to the homozygous PTM group either. This should be considered because if the sample size was larger, it could not be ruled out that patients with heterozygous PTM too could have insufficient anticoagulation with dabigatran etexilate. We also did not measure thrombin or prothrombin plasma levels.

Although thrombin plasma levels are not the only explanation and reason for hypercoagulability and a higher risk of thrombotic events in factor II mutations [37], it would have been a very interesting parameter.

### Further Studies Must Investigate the Efficiency of Dabigatran Etexilate in Patients with PTM

Due to the high rate of PTM (up to 30%) in CSVT and the increasing use of dabigatran etexilate, we believe genetic screening for thrombophilia in those patients should be performed to allow patient tailored therapeutic decisions regarding oral anticoagulation. Thereby, genetic screening may prevent treatment with insufficient anticoagulation and recurrent thrombotic events. We recommend using dabigatran etexilate with caution in homozygous PTM.

## 5. Conclusions

Our project was the first study that, to our knowledge, showed therapeutic dabigatran peak concentrations in patients with CSVT and some types of genetic thrombophilia. None of our patients with sufficient dabigatran peak concentrations had any recurrent thrombotic events for 24 months, which suggests that dabigatran is efficient in CSVT and heterozygote FVL, but should be treated with care in PTM.

## Figures and Tables

**Figure 1 life-12-00970-f001:**
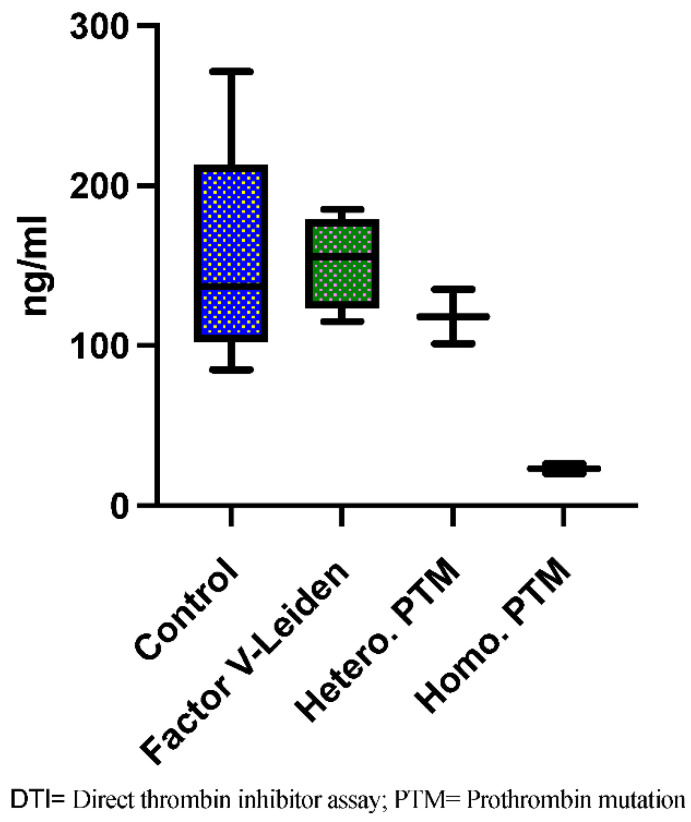
DTI peak at BL.

**Figure 2 life-12-00970-f002:**
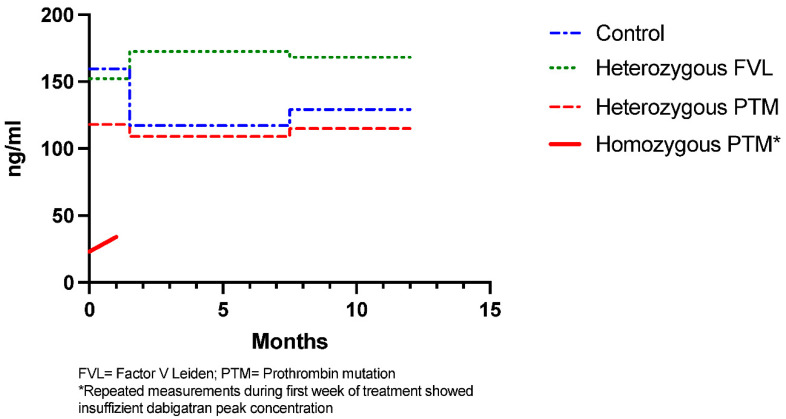
Longitudinal dabigatran peak concentration.

**Table 1 life-12-00970-t001:** Demographic and laboratory at baseline.

	All Patients *n* = 24		Genetic Thrombophilia (*n* = 10)		Without Genetic Thrombophilia (*n* = 14)		*p*-Value
	Mean ± Std	Median	Mean ± Std	Median	Mean ± Std	Median	
Age	55.4 ± 19.3	58.5	48.5 ± 14.92	50	60.3 ± 21.2	69.5	0.1
BMI	27.3 ± 4.7	27	26.7 ± 4.3	27	27.6 ± 4.6	28	0.67
Creatinine Clearance	0.89 ± 0.27	0.86	0.89 ± 0.29	0.86	0.90 ± 0.28	0.92	0.92
GOT/AST	22.6 ± 6.8	22	20.5 ± 7.1	21	24.1 ± 6.4	23	0.33
GPT/ALT	22.5 ± 7.8	20.5	19.9 ± 6.3	19.92	24.4 ± 8.5	23	0.36
Dabigatran peak concentration (ng/mL)	141.0 ± 60.7	130.5	119.3 ± 52.3	130.5	159.6 ± 63.1	137	0.29
TT (s)	147.0 ± 29.5	155.5	139.4 ± 43.9	161.0	152.5 ± 11.1	153	0.5
APTT (s)	37.8 ± 6.8	36.2	35.9 ± 43.9	35.6	39.1	39.6	0.19

BMI = Body Mass Index; GOT/AST = aspartate transaminase plasma levels; GPT/ALT = alanine transaminase levels; TT = Thrombin time; APTT = activated partial thromboplastin time.

**Table 2 life-12-00970-t002:** Variant genetic disorders observed and dabigatran peak concentration.

	*n* (%)	Dabigatran Peak Concentration ng/mL (Mean ± Std)
Homozygote PTM	2 (20)	23 ± 4.2
Heterozygote PTM	2 (20)	118 ± 24
Homozygote FVL	0 (0)	-
Heterozygote FVL	7 (70)	152.3 ± 27.5
Homozygote PAI-1 4G/5G polymorphism	2 (20)	23 ± 4.2
Heterozygote PAI-1 4G/5G polymorphism	3 (30)	123 ± 32.3
Multiple genetic disorders *	5 (50)	70.5 ± 56.6

FVL = Factor V Leiden; PTM = Prothrombin mutation. * 1 patient: homozygote FII and homozygote PAI-1 4G/5G polymorphism; 1 patient: homozygote FII mutation, homozygote PAI-1 4G/5G polymorphism and heterozygote FV mutation; 3 patients: heterozygote FV mutation and heterozygote PAI-1 4G/5G polymorphism.

## Data Availability

Data available on reasonable request.

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
