# Peer review of "Dabigatran in Cerebral Sinus Vein Thrombosis and Thrombophilia"

_life, 2022, doi:10.3390/life12070970_

Round 1

Reviewer 1 Report

The authors discuss dabigatran peak concentration is stable with heterozygote FVL and heterozygote PTM, but not in homozygote PTM compared to controls. The effectiveness of drugs has been illustrated from a novel perspective. But there are still some flaws.

1.   Inclusion and exclusion criteria need to be further clarified.

 2.     How Patients with CSVT who also suffer from prothrombin-mutation (PTM) or Factor V Leiden (FVL) are diagnosed. This requires further clarification in the Methods section.

 3.     Author mentioned all patients reported regular intake of 150 mg dabigatran etexilate twice a day. What is the basis for drug dosage?

4.     It is not appropriate to start a sentence with a numeric. For example, 2 out of the 10 patients with acute CSVT and genetic thrombophilia had multiple insufficient dabigatran peak concentration (26 and 20 ng/ml). 2 should be two.

 5.     What is the basis for diagnosis of acute CSVT, please specify.

 6.     What is the reason why APTT and TT are selected among the commonly used coagulation parameters, while other parameters are not selected, such as PT, platelet and something like that.

 7.     Additional English correction is not required.

Reviewer 2 Report

Dear authors,

Congratulations for this useful paper! I would like to give some comments on your work:

Line 101- please explain the acronym, EHRA, in your text

Line 105- which database did you use concretely?

Line 114- why is the reason to check the peak plasma concentration after 48 h?

Line 158: dot after fibrillation).

Line 256: 2 dots after dabigatran etexilate!

Line 286: Please correct the font for Bristol - Myers Squibb

I suggest for discussion to refer also to this  3 important paper:

Bose G, Graveline J, Yogendrakumar V, et al-Direct oral anticoagulants in the treatment of cerebral venous thrombosis: a systematic review BMJ Open 2021;11:e040212. doi: 10.1136/bmjopen-2020-040212

Khider, L.; Gendron, N.; Mauge, L. Inherited Thrombophilia in the Era of Direct Oral Anticoagulants. Int. J. Mol. Sci. 2022, 23, 1821. https://doi.org/10.3390/ ijms23031821

Nepal G, Kharel S, Bhagat R, Ka Shing Y, Ariel Coghlan M, Poudyal P, Ojha R, Sunder Shrestha G. Safety and efficacy of Direct Oral Anticoagulants in cerebral venous thrombosis: A meta-analysis. Acta Neurol Scand. 2022 Jan;145(1):10-23. doi: 10.1111/ane.13506. Epub 2021 Jul 21. PMID: 34287841.
